# A Comprehensive Survey on Affinity Analysis, Bibliomining, and Technology Mining: Past, Present, and Future Research

**Md. Rashadur Rahman** [1], **Mohammad Shamsul Arefin** [1,2], **Sanjida Rahman** [1], **Afsana Ahmed** [1], **Tahsina Islam** [1], **Pranab Kumar Dhar** [1] **and Oh-Jin Kwon** [3,*]

1   Department of Computer Science and Engineering, Chittagong University of Engineering and Technology, Chattogram 4349, Bangladesh; rashadur@cuet.ac.bd (M.R.R.); sarefin@cuet.ac.bd (M.S.A.); sanjidadia10@gmail.com (S.R.); afsana.ahmed1000@gmail.com (A.A.); tahsinaislam385@gmail.com (T.I.); pranabdhar81@gmail.com (P.K.D.)
2   Department of Computer Science and Engineering, Daffodil International University, Dhaka 1341, Bangladesh
3   Department of Electrical Engineering, Sejong University, 209 Neungdong-ro, Gwangjin-gu, Seoul 05006, Korea
*   Correspondence: ojkwon@sejong.ac.kr

**Abstract:** Recent advancements in high-speed communications and high-capacity computing systems have contributed to major growth in the data volume of databases. Data mining is a crucial part of information retrieval; it is often termed as database knowledge discovery. It consists of techniques for examining massive data sets, to find hidden (but possibly important) information. Three interesting fields in data mining are affinity analysis, bibliomining, and technology mining. Affinity analysis provides data mining techniques to determine the similarity among objects; bibliomining is a combination of data mining, bibliometrics, and data warehousing; technology mining is a research topic that is an obstacle to many scientists in the fields of time association, enterprise association, and computer programming. We present a systematic review of the notable research articles in the fields of affinity analysis, bibliomining, and technology mining published between 2000 and December 2021. We provide a systematic analysis of the selected literature by specifying the major contributions, used data sets, performance evaluations, and limitations. Our findings demonstrate that affinity analysis interoperability extends well beyond market basket analysis. We also demonstrate that, in the age of big data, the personalized needs of users are the driving forces behind the evolution of the digital library from a resource-sharing service to a user-centered service. Finally, this article provides insight into major advances and outstanding challenges in the fields of affinity analysis, bibliomining, and technology mining.

**Keywords:** affinity analysis; association mining; correlation mining; bibliomining; technology mining

## 1. Introduction

In science, research findings and phenomena are expressed using data. Data allow for the recognition of efficiency improvement and optimization opportunities in manufacturing, diagnosis, prediction, and other fields, as well as the optimization of troubleshooting procedures. The massive expansion of data has provided an impetus to develop a means to identify significant patterns within vast amounts of information. Advanced data collection technologies and information management systems have expanded our capabilities to capture and efficiently store data of all sizes, in line with the exponential rise in the volume and sophistication of data. Frequent pattern mining is a major area of data analysis; it has been investigated through a wide range of applications, such as association rules [1–4], closed frequent patterns [5–8], constraint-based frequent pattern mining [9–11], and correlations [12–15]. Our capacity to view and evaluate data, however, is still limited; there is an immediate need to speed up the exploration of data in databases.

### 1.1. Affinity Analysis

An affinity analysis refers to the study of when two or more items are related to one another [16]; as such, affinity analysis provides data mining techniques that are used to determine similarities among objects [17]. The similarities could be between users of an e-commerce site (e.g., in order to provide appropriate advertising and targeted services), between items that are to be sold to the users, or between human genetics to identify individuals that have the same ancestors. An affinity analysis can be performed to explore the inherent relationships, correlations, and regularities in a data set [18,19]. As such, an affinity analysis can be extended to procedures where agent recognition is specific and it is possible to record details about operations. In fact, an affinity analysis is primarily used for market basket analysis [20] that marketers conduct to learn about consumer behaviors in relation to the transactions they produce. The first phase in the study of an affinity analysis is recognition of the subject, which may be identified under certain circumstances. The next step is to observe and then report on the subject's behaviors. A pattern may appear upon recording, which can then be used to create correlations. In marketing and sociology, an affinity analysis is largely used to explain similarities between incidents that are linked but not specifically related. Although an affinity analysis is a promising field of data mining, to the best of our knowledge, there are no reviews focused on the affinity analysis. In this study, we provide a review of the existing approaches to an affinity analysis in various applications.

### 1.2. Bibliomining

The term bibliomining was first utilized by Scott Nicholson and Stanton in 2003 to discuss data warehouses for libraries, the goal of which was to exploit the informal communities that power both bibliometric- and client-based data mining through a solitary data warehouse [21,22]. Bibliomining can be utilized to determine the patterns in the items that individuals are perusing and investigating; it permits administrators to better focus on their locales. It can, likewise, help library directors to center their spending plans around assets that will be used. Bibliomining for libraries involves the use of data mining and bibliometric techniques on information created from library administrations [23,24]. It depends on the applications of data mining methods in libraries, in order to extract data that facilitate the dynamic (or more prominent) effectiveness of administration. Many studies have explored bibliomining as it has increased in popularity. Various approaches for arranging and cleaning information before warehousing have been proposed; different central zones for investigation are increasing. Bibliomining has even been performed on surveys dependent on library utilization, where circulation information is absent. Each bibliomining investigation can uncover patterns, identified through various kinds of exercises in libraries. The most important contribution has been described by Nicholson; according to [22,25–27], the bibliomining cycle is comprised of:

1. Determining the areas of focus;
2. Identifying the internal and external data sources;
3. Selecting appropriate analysis tools;
4. Discovering patterns through data mining;
5. Analyzing results.

Bibliomining, otherwise called data mining for libraries, provides many instruments that can be utilized to determine patterns within a volume of crude information, and can deliver the examples expected to form a model for automated assortment advancement [28]. One objective of this paper was to investigate the idea of bibliomining. Initially, the roots of the term and the relationships between bibliomining and its significant parts—bibliometrics and information mining—were investigated.

### 1.3. Technology Mining

Technology mining (or tech mining), is one of the most promising fields of study in the technology management sector. Over the past few decades, this area has been fruitful for

many researchers in the areas of organization, enterprise organization, and software engineering branches. Most technology mining programs focus on licenses, patents, and R&D reports [29]. Other researchers in the innovation mining field have characterized 'tech mining' in their studies [30], showing that it can be utilized as a technique to help small firms with low proficiencies to form cutting-edge considerations. As demonstrated in [31], the degree of distributed papers and the degree of references in technology mining-related journals have exhibited progress; there was a significant increase in the number of distributions after 2005 and a major climb in the number of references after 2012. Emerging expertise is driven by innovative work and by customer and corporate interests, in terms of new items and applications.

In this paper, we present a comprehensive review of the above-mentioned topics, based on different application domains. We provide a comprehensive review of the relevant literature by identifying the major contributions, data sets used, and performance evaluations therein. The primary purpose of this work was to provide a considerably better understanding of the existing works, their limitations, research gaps, and future research directions in the fields of affinity analysis, bibliomining, and technology mining. We first identified some research gaps in the previous surveys, then structured several research questions to facilitate our research hypothesis. Our review results indicate that the use of an affinity analysis beyond association rule mining as well as the personalized needs of users are the driving forces behind the evolution of digital libraries. This research may be considered an excellent resource for interested researchers, practitioners, and readers, as it provides insight into the significant advances, challenges, and research directions associated with the fields of affinity analysis, bibliomining, and technology mining.

### 1.4. Organization

The remainder of this paper is organized as follows. The related literature reviews are discussed in Section 2. Section 3 describes the methodology of our systematic review. The distributions of the publications selected for review are shown in Section 4. The review of the affinity analysis is discussed in Section 5, while the reviews of bibliomining and technology mining are discussed in Sections 6 and 7, respectively. Section 8 provides the answers to the research questions, and details challenges, open issues, and future research directions. Section 9 concludes the review with a summary.

## 2. Related Literature Reviews

Review works on association mining, itemset mining, market basket analysis, and data mining can be found in the literature. This section presents review papers for the mentioned fields, with their respective factors and shortcomings.

The task of discovering correlations between elements in a data set is known as association rule mining. Ceglar et al. [32], in their review, focused on the basic concepts of association mining such as itemset identification, rule creation, and general optimizations. In this review, we organized the significant essential contributions made in association mining research, emphasizing the development of the underlying concepts regarding item identification and rule generation. Market basket analysis is one of the most-used methods in data mining, which is used to examine the purchasing behaviors of customers. It is a method used for determining the links or associations between the various goods in a customer's cart. Maske et al. [33] discussed the various approaches for market basket analysis. Most of their primary studies focused on using the Apriori algorithm for generating frequent itemsets; they did not describe other algorithms for market basket analysis in detail.

High-utility pattern mining is a new data science approach that involves finding patterns with high significance in a data set. A pattern's usefulness may be assessed using a variety of objective measures, such as profit, frequency, and weight. Viger et al. [34] presented an overview of high-utility itemset mining. They reviewed the state-of-the-art algorithms, expansions, and applications for high-utility itemset mining; furthermore, they

examined the basic strategies for exploring the search space of itemsets used in high-utility itemset mining algorithms and presented the problems associated with high-utility itemset mining. Finally, they proposed several research opportunities, such as enhancing the performance and extending the algorithms to incorporate more complex data patterns, as well as applications in novel fields.

Correlated high-utility pattern mining (COHUPM) is an extension of high-utility pattern mining. COHUPM uses both utility and correlation measurements to uncover intriguing high-utility patterns. The COHUPM approaches in the literature were studied and compared by Almoqbily et al. [35], who reviewed the most up-to-date methodologies, critical metrics, data structures, and pruning features in great detail. Their work highlighted the various data structures used in COHUPM techniques and discussed different pruning properties. They also pointed out the necessity of new and efficient data structures and pruning properties for better performance. In addition, they discussed the present difficulties inherent to COHUPM techniques and future research directions. The newest research on incremental high-utility itemset mining algorithms was presented by Cheng et al. [36]; it included techniques for storing itemsets and utilities using tree, list, array, and hash set storage structures.

There are very few review papers available in the field of bibliomining. Siguenza et al. [37] presented a thorough assessment of the literature as well as a categorization system for data mining techniques used in academic libraries. A total of 41 practical contributions from 1998 to 2014 were selected and evaluated for direct relevance. They categorized each article into four primary data mining functions: classification, regression, association, and clustering. The main goal of the literature review of Barashdi et al. [38] was to determine the approaches and technologies that are used in academic libraries to analyze big data, and to see how much profit could be made from them. Their study provided basic knowledge regarding the benefits of big data in academic libraries and future research prospects.

Hamad et al. [39] investigated the concept of big data from the perspective of information technology department staffs at three major university libraries in Jordan. Another article [40] examined and anticipated data mining trends and hotspots in Chinese academic libraries. The scope of this study was limited to the China National Knowledge Infrastructure (CNKI).

*Research Gaps*

Considering the above-mentioned approaches, some notable research gaps are listed below:
- There are no existing reviews dedicated to affinity analysis;
- No approaches exist for reviewing affinity analysis-based research papers, according to their application domains;
- No effort has been made to review technology mining literature.

## 3. Review Methodology

A systematic literature review (also known as a systematic review) is a method used for discovering, analyzing, and interpreting all research relevant to a specific research question, topic area, or phenomenon of interest [41]. A systematic review is an effective tool used for obtaining an overall concept of the prior research works performed and their related results in a particular domain. A systematic review provides a method for locating, selecting, and critically assessing papers that address specific research topics. When conducting a systematic review, several guidelines needed to be followed [42]. The objective of our review was to impart a detailed review of the notable research works performed in the mentioned domains, along with their key contributions. We developed a structural format based on characterization features for our review, using the directions presented by [43,44]. The major steps we followed when conducting our review are described in the following subsections.

### 3.1. Analyses of Related Surveys

Analyses of the related literature reviews are provided in Section 2.

### 3.2. Definition of Research Questions

The selections of primary studies were based on whether or not the study could assist in answering our research questions. By evaluating existing studies, we aimed to answer the following research questions (RQs).

- RQ1: what are the different types of application domains of affinity analysis?
- RQ2: how are affinity measures incorporated to improve itemset mining?
- RQ3: what are the key bibliomining practices and technologies available for academic libraries?
- RQ4: what are the existing techniques and approaches for technology mining?
- RQ5: what are the existing open issues, key challenges, and future research directions in the fields of affinity analysis, bibliomining, and technology mining?

### 3.3. Identification of Studies

One major step involved determining the primary data sources from which the primary articles were gathered. We used Google Scholar as the primary search engine for the selection of primary studies. To find relevant articles, we also considered the various renowned academic publishers, including Scopus, IEEE, SpringerLink, MDPI, Hindawi, Wiley, Taylor and Francis, ACM Digital Library, ScienceDirect, and Emerald.

We used various search strings to search for research articles in the intended domains. To explore articles related to the affinity analysis, we used the following search strings:

*(affinity <OR> correlation)*
*[AND]*
*("association rule" <OR> weight <OR> set <OR> association)*
*[AND]*
*(analysis <OR> mining).*

For bibliomining, we used the search string:

*bibliomining <OR> bibliometrics <OR> "smart libraries".*

Finally, we used the following search string to search for notable research articles in the technology mining field:

*( technology )*
*[AND]*
*( evolution <OR> associative <OR> " " )*
*[AND]*
*(analysis <OR> mining <OR> network <OR> ecosystem ).*

### 3.4. Study Selection

The selection of studies was carried out in two phases: primary selection and final selection.

#### 3.4.1. Primary Selection

We initially selected the primary sources based on titles, keywords, and abstracts; however, in the cases where the titles, keywords, and abstracts gave inadequate information, the evaluation was expanded to include the conclusions section. This stage yielded about 1073 publications, including conference papers, journals, posters, summaries, book chapters, books, symposiums, and other literature.

#### 3.4.2. Final Selection

The potential of a research article was assessed based on various criteria, including the scope of the research, key contributions, and future research impacts. The inclusion/exclusion criteria used to include relevant articles and eliminate irrelevant ones are

shown in Table 1. To be selected for our review, an article had to satisfy both inclusion criteria $InC_1$ and $InC_2$, as well as any one of the following: $InC_3$, $InC_4$, or $InC_5$. An article that met any of the exclusion criteria was not considered for further review.

**Table 1.** Inclusion and exclusion criteria for determining relevant papers.

| ID | Inclusion Criteria |
|---|---|
| $InC_1$ | The article is published in conference proceedings or in a journal. |
| $InC_2$ | The main contribution(s) of the study focus on affinity analysis, bibliomining, or technology mining. |
| $InC_3$ | The article is published between the year 2000 and December 2021. |
| $InC_4$ | The article proposes at least one affinity analysis method or applies any affinity analysis method to solve any real-world problem. |
| $InC_5$ | The article proposes at least one method for bibliomining or applies any method to solve any real-world bibliomining problem. |
| $InC_6$ | The article proposes at least one method for technology mining or applies the method to solve any real-world problem related to technology mining. |
| | **Exclusion criteria** |
| $ExC_1$ | The article is a poster, editorial, report, summary, or book review. |
| $ExC_2$ | The article is not in the English language. |
| $ExC_3$ | The main focus of the study is not relevant to affinity analysis, bibliomining, or technology mining. |
| $ExC_4$ | The major contribution of the study is unclear. |
| $ExC_5$ | Duplicate study or article. |
| $ExC_6$ | The article is unavailable in electronic format. |

### 3.5. Data Extraction and Synthesis

After selecting appropriate primary studies for review, the next task was to perform an in-depth study of each selected paper. The key objective of this review was to identify the significant contributions, implementations, evaluation criteria, and data sets used in each paper. The distribution of the selected primary studies is presented in Section 4. The extracted relevant data from the primary studies are presented in Sections 5–7. The information extracted from the primary studies was then synthesized to answer our research questions, as described in Section 8.

## 4. Publication Distribution

Selecting quality research articles is a vital task when writing a survey paper. Not all of the published research articles in any particular domain are high-quality. We selected notable research articles from renowned publishers published in six different time frames, choosing the time frames in such a way that our survey covered the latest research as well as earlier research works in the domains of affinity analysis, bibliomining, and technology mining. The readers can see a sequential overview of the works conducted in different time frames in Table 2.

**Table 2.** Time distributions of the papers considered for review.

| Before 2005 | 2006–2010 | 2011–2012 | 2013–2015 | 2016–2018 | 2019–Dec 2021 |
|---|---|---|---|---|---|
| Shyu et al. [45] | Yun et al. [46] | Wang et al. [47] | Li et al. [48] | Rezgui et al. [49] | Tshimula et al. [50] |
| Changchien et al. [16] | Yun et al. [51] | Panigrahy et al. [52] | Li et al. [53] | Hong et al. [54] | Butt et al. [55] |
| Xiong et al. [56] | Wang et al. [57] | Subbiah et al. [58] | Masum et al. [59] | Moraes et al. [60] | Li et al. [61] |
| Shyu et al. [62] | Chen et al. [63] | Liu et al. [64] | Zhang et al. [65] | Sadiq et al. [66] | Srilatha et al. [67] |
| Shen et al. [68] | Li et al. [69] | Hajek et al. [70] | Azam et al. [25] | Zhang et al. [71] | Swofford et al. [72] |
| Nicholson et al. [28] | Pitkaranta et al [73] | Hajek et al. [23] | Deshpande et al. [27] | Chen et al. [74] | Yadav et al. [75] |
| Nicholsen et al. [22] | Nicholson [21] | | Madani et al. [31] | | Bao et al. [76] |
| Prakash et al. [77] | Nicholson [78] | | | | Wen et al. [79] |
| Kostoff et al. [80] | Shieh et al. [24] | | | | Zeba et al. [81] |
| Kao et al. [82] | Nicholson et al. [26] | | | | |
| Wu et al. [83] | Courseault et al. [30] | | | | |
| Yoon et al. [84] | Adomavicius et al. [85] | | | | |
| Porter et al. [29] | | | | | |

We selected 29 papers on affinity analysis, 16 papers on bibliomining, and 8 papers on technology mining from various publishers, including IEEE, ACM, ScienceDirect, Springer-

Link, Wiley, and others. The distributions of the 53 selected publications, according to data sources, are shown in Figure 1.

**Figure 1.** Distributions of the publications according to data sources.

## 5. Affinity Analysis Review

Queries aim to access information from media objects that are structurally similar and that exist in various databases, contributing to a need to explore systemic equivalence links throughout the databases (Table 3). A generic affinity-based association rule mining technique for network databases was proposed in [45] for the discovery of quasi-equivalence relationships. There are two phases in the proposed generalized affinity-based association mining procedure. In the first phase, a refinement and interest constraint, as well as minimum interest thresholds, are iteratively checked. In the second phase, a minimum confidence threshold limit is tested; if an unacceptable scenario occurs, additional conditions may also be applied. In order to measure the frequency of two objects that have been accessed together in a set of queries, a relative affinity value between the two objects is utilized. Here, the query set is known to be a transaction set, as each query may request information from one or more database media objects (similar to how each transaction must contain one or more items). They calculated the confidence, interest, and support of their association rules based on mutual affinity values. Using relative affinity metrics gives more insightful feedback, as they indicate the number of query accesses, but not the number of queries. The proposed methodology has been evaluated on the Purdue University database management system (i.e., a real database system). Their experimental results showed that the proposed generalized approach for affinity-based association rule mining could properly extract a set of media objects that are quasi-equivalent from the database. Their proposed mechanism performs better than the basic association rule mining methodology in discovering pairs of quasi-equivalent media objects.

To prune the combinatorial search space, current association-rule mining algorithms frequently rely on support-based pruning approaches. This technique is not very practical for data sets with distorted support ratios, as they appear to produce several spurious trends involving objects from various support levels or may skip potentially interesting low-support patterns. Xiong et al. [56] suggested using a hyperclique trend to solve these issues; they used an analytical metric called h-confidence to define clear patterns of affinity. To avoid behaviors involving objects with significantly different support levels, they also introduced the modern idea of the cross-support property. They suggested a new hyperclique miner algorithm that utilizes h-confidence cross-support and anti-monotonic properties to successfully discover hyperclique patterns. Later, hyperclique pattern applications were shown to have good ability to detect strong affinity patterns among low-support objects and cluster hyperclique-based items. Their experimental

findings indicated the efficacy of this approach for identifying patterns even at very low support levels in dense data sets, in which most current algorithms break down.

Yun et al. [46,51] suggested a unique algorithm, named the weighted interesting pattern mining (WIP), which applies a new metric of weight-confidence (w-confidence) to weight affinity patterns associated with mining. WIP aims to utilize weight trust and/or h-confidence in the weighted frequent pattern mining algorithm, based on the approach to pattern development, in order to prune uninteresting patterns. In order to determine weight limits, a weight range is used and an h-confidence score allows for the recognition of strong support affinity patterns. The proposed WIP algorithm balances the two measures of weight and support. The performance was improved by incorporating h-confidence and a modified FP-tree structure without the additional cost of computing. Pumsb, Mushroom, and Connect are the three real data sets used for evaluating the proposed algorithm. Experiments with these real data sets demonstrated that WIP performs better in the case of weighted frequent pattern mining. Another advantage of WIP is that it produces fewer, but more useful patterns. The proposed WIP algorithm is a pattern mining algorithm based on the main memory; as such, this approach might be impractical when the size of the data set is very large or when a minimal minimum threshold is used. This work can be upgraded to a disk-based mechanism.

Yadav et al. [75] proposed a scalable Hilbert–Schmidt independence criterion (HSIC), a probabilistic dependency-based metric, in order to measure the proximity between data points for graph learning. They verified the effectiveness of this technique using both synthetic and real-world data sets. Experiments with the HSIC-based affinity produced outstanding results, showing that it is capable of determining the relationships between data points and precisely obtaining the associated graph. It could also assess the genuine similarities between data points on synthetic data sets. Similarly, its overall accuracy on real-world data sets is much greater than that of existing techniques, indicating that the affiliations derived by HSIC provide superior representations of the similarities between data points.

### 5.1. E-Commerce

Changchien et al. [16] proposed a mechanism for mining association rules for online e-commerce recommendations. For this purpose, they attempted to infer the hidden habits of customers for product recommendations. The proposed framework consists of two major phases. In the first phase, affinity grouping is performed for a large number of records. This phase is called the clustering module. Clustering is performed based on a self-organization map (SOM) and an artificial neural network (ANN). The next phase is to derive association rules in each cluster and find relations between different clusters using a rough set. The ideas of lower and upper sets are included. They divided the clients, goods, and business vendors into clusters by SOM.

The work by Bao et al. [76] provides a novel bi-directional affinity measure framework. They conducted an experiment on a groceries data set to assess whether the bi-directional measuring approach of frequent patterns could function well in large data volumes. The IBM SPSS Modeler data set (1000 shopping records and 11 items) was also used to test the coherence and consistency of the suggested scheme. The evaluation results indicate that the novel association rule measurement approach provides excellent accuracy and stability for evaluating and determining association rules. It can also be utilized in business settings for paired recommendations.

The authors of [57] focused on the analysis of market basket data; they found affiliations among items within large groups of transactions. They introduced the weighted confidence (WC) affinity function to assess the resemblance between clusters. This method is non-parametric, as the user does not need to provide any input parameter, and each item is represented using a bit vector to employ the vertical data structures, such that utilizing logical operations to speed up the clustering process is possible. After an evaluation, it

was reported that the WC-clustering method can determine the proximity between clusters more effectively than other affinity measures.

In the paper [47], the authors came up with a solution for grouping classified data such as market basket data sets. They proposed a parallelized hierarchical clustering technique for categorical data. This approach is hierarchical and, hence, non-parametric (i.e., the user does not need to provide input parameters); it was implemented using MPI. This clustering approach represents each item as a bit vector to utilize vertical data structures, and applies logical operations to accelerate the computation of itemset support and the merging process. Experiments were conducted on standard benchmark market basket data sets, and the results demonstrated that the proposed PH-clustering method outperforms the sequential approach drastically without compromising the clustering quality; furthermore, this acceleration has a significant impact in large-scale data sets.

Srilatha et al. [67] presented a new extraction technique for cumulative database systems, which may be used to determine the high-utility itemsets that have been uncovered. To find a highly profitable itemset with a significant frequency affinity, in each transaction, affinity utility values were used instead of high-utility values. In addition, a knowledge-weighted utilization (KWU) upper bound was used to preserve the downward closure characteristic. Their experimental results revealed that the recommended utility mining strategy has improved the runtime and memory utilization when compared to other conventional utility mining methods, such as LIHUP and FUP-HAUIMI, in a cumulative database.

*5.2. Social Network Analysis*

Panigrahy et al. [52] introduced sketch-based novel affinity measures in social networks, mainly considering two properties—the space between two users and the total edge-separated paths between them—to determine their joint robustness. An experimental study was developed by evaluating two data sets: one was a Hotmail email exchange graph that dealt with population data and the history of a Bing search, and the other was a Twitter mention-graph with underlying tweet text. By comparing this affinity measure to the approximate shortest path distance, it was perceived that higher-affinity users are related to each other regarding kinship or distance, and that there is high resemblance regarding queries, demographics, and tweets.

Li et al. [53] proposed a generative graphical model (namely, the UCT model) to determine the context of social media data that are concurrently user- and community-based. According to the model assumption, they proposed one more structure, FRec, which can develop relevant user and community recommendations for a target user. They conducted an extensive experimental evaluation on a data set; their results showed that: (1) this statistical model gives high-quality results, compared to various standards and typical topic models, while pointing out user and community issues related to generalization errors and computational complexity; and (2) the proposed framework offers more logical recommendations for users and communities than other schemes. This framework has been conveniently integrated into real-life social networks, due to sophisticated technology that minimizes training time and recommendations during training.

Rezgui et al. [49] proposed a system, named "Affinity Finder", in order to discover hidden affinity relationships between Twitter users. The proposed system also evaluates tweets to extract association scores that affect the degree of affinity among Twitter users. After the evaluation, it was found that this system can effectively accumulate, keep, and operate many tweets. After clarification, the system was found to be competent in managing vast amounts of relationships, and can analyze the results entirely within a short time. This system can provide necessary data to target users and companies.

In [50], the issue of finding hidden affinity associations within the internet was addressed. The authors proposed an algorithm, named HAR-search, to detect hidden affinity relationships between individuals. HAR-search derives positive interaction sequences (PIS) regarding the context of messages in the discussion history. Markov chain models are used

to assess the PIS with respect to affinity scores. These values refer to the degrees of affinity that users possess. After evaluating a real data set, a considerable difference was observed with respect to the comparative method, regarding the list of recorded associations. Moreover, a HAR-search could reveal many more associations on the experimental data sets than the Affinity Finder.

Blog Analysis

Li et al. [69] elaborately analyzed how information is circulated in the blogosphere by considering the blog cascade affinity. Their proposed method identifies and analyzes a group of properties that may influence the tendency of bloggers to join a cascade and it utilizes these properties to predict the cascade affinity of blogs. They conducted an ANOVA test on these properties and found that, apart from the citation factor, the rest of the properties have powerful impacts on the cascade proximity. The cascade affinity prediction was then defined as a classification task and an SVM classifier was employed for the experiments. Utilizing the SVM prediction scores, candidate blogs can feasibly join a cascade. Their test identified that, of all the properties, the number of friends plays the most significant role in predicting the blog cascade affinity.

The work in [48] focused on identifying and analyzing a group of content-oblivious properties (both large- and small-scale) that induce a blog's tendency to enter into a cascade. Based on these features, the authors presented two schemes—a support vector machine (SVM) classification-based approach and a bipartite Markov random field-based (BiMRF) scheme—to assess the feasibility of a blog's proximity to a specific cascade. Then, the blogs were sorted according to feasibility. Working on a real data set, this test identified that, of all the properties, the number of quasi-friends plays the most crucial role in predicting the cascade proximity of blogs. The proposed prediction scheme can generate much more accurate results than other schemes using only conventional properties.

*5.3. Natural Language Processing*

Many clustering algorithms have been developed and thoroughly researched in the record retrieval field, in order to increase the efficacy and performance of retrieval. The authors in [62] recommended an affinity-based similarity measure approach that incorporates user access patterns using a probabilistic model to assess the correlations between web documents. Their suggested affinity-based probabilistic algorithm, which adopts the Markov model principle, is used to dynamically mine the similarity of documents based on user access patterns. User access patterns are derived from server log records, which reflect the complex characteristics of web documents in the sense that, by following the hyperlinks provided in the web documents, each user has browsed the web site. For validation, the Euclidean distance, Cosine coefficient, and the proposed affinity-based measure were applied in four clustering algorithms: complete link, partitioning around medoids (PAM), group average link, and single link. The results showed that the proposed affinity measure, which utilizes only the static information in the data, produces far better clustering results than the other two clustering measures.

The authors of [60] considered the problem of the context-free grammar (CFG) similarity in order to point out the constructional affinities between languages. The authors presented a simple and efficacious algorithm to analyze the CFG affinity; they evaluated its potential and efficiency. No strategy had previously been proposed to analyze context-free grammar affinity; as such, they aimed to alleviate that limitation using the proposed method. Moreover, this method gives the most proficient (and relevant) solution to the problem of CFG similarity. This approach was also intended to be used in building a system to extract organized data items, such as music, which can be perfectly exhibited using CFG.

The increase in popularity of social media platforms, such as Facebook, Twitter, and Snapchat, has been hampered by an influx of undesired and disturbing characters on these platforms. Sadiq et al. [66] introduced a novel scheme, known as the associative affinity factor analysis (AAFA), for stance detection and identification of bots. The suggested

approach uses AAFA to distinguish real people from bots and to detect the stance in bipolar affinities. The 2016 U.S. election campaign tweet data were used for their proposed model. Moreover, machine learning algorithms were applied for stance analysis in election predictions for the first time. This framework can correctly determine the reasons underlying the number of Twitter followers and social media fame by differentiating authentic followers from paid bots. After the evaluation, it was found that the correctness of the proposed framework was very high, compared to other prominent tools in the industry.

### 5.4. Video Data

Masum et al. [59] proposed a methodology for finding kinship in a vast amount of video-based data sets. Alleged attachments from video surveillance trajectories have potential applications for improving human social interactions, early detection of potential events, and even tracking accuracy. In order to easily determine the presence of proximity, they introduced a supervised learning technique, which could be used for further assumptions if required. To point out the similarity, proximal signals are used to acquire affection-associated properties. The PCA-based dimensionality-reducing scheme and the cosine angle-based affinity comparison schemes were employed to discover the presence of similarity in a pair of subjects. After the evaluation, this assumption-based affinity identification scheme showed optimal accuracy in kinship identification when compared with the ground truth.

Hong et al. [54] introduced a method that mainly focuses on the closeness between individuals, depending on their movie-watching histories. This recommendation scheme is made up of (a) the computation of a similarity metric between movies, according to the weighted properties; (b) the creation of a fundamental affinity network graph; and (c) the calculation of a user's proximity to a group, based on the graph. To evaluate the proposed social affinity-based group movie recommendation (SAGRS) method, virtual users with movie histories were created. After evaluating the synthetic data set, it was found that the proposed scheme can efficiently discover social affinities.

For conversational group recognition, Swofford et al. [72] presented a new approach called DANTE. To discover team cohesion based on visual cues of spatial activity, this method integrates graph aggregation with recent deep learning algorithms. The proposed approach outperformed earlier methods in several conversational group identification benchmarks. It also obtained good results under the GDSR metric in more general group identification tasks. Furthermore, using data-driven methods allowed this scheme to deal with complex spatial patterns of behavior without the need for ad hoc measures to validate team cohesion. These properties make this scheme stable and pragmatic enough to be used in a genuine human–robot interaction context.

### 5.5. Healthcare

An affinity set is a data mining method that can be used to identify, evaluate, and construct the relationship between a knowledge system's observed effects (consequences) and potential incomes (causes). Chen et al. [63] proposed a mechanism based on an affinity set to classify core characteristics contributing to delayed medical diagnoses. They gathered data from a hospital emergency room (ER), with the aim of identifying key attributes that could contribute to the issue of delayed diagnosis through the data mining affinity collection. They discussed the delays in detecting or breaking down scheduled operational interventions in order to reduce the likelihood of errors. Diagnosis delay refers to those medical injuries in the emergency department (ER) that are overlooked or skipped, but which physicians in the intensive care unit recognize. The efficiency of the proposed model based on the affinity set was contrasted with the rough set model, SVM, logistic regression, and a neural network. The model using affinity demonstrated its benefits, in terms of precise estimation and power of explanation.

### 5.6. Affinity Propagation

Affinity propagation has been applied successfully in several fields, such as biometric identification. In contrast, Liu et al. [64] implemented an affinity propagation algorithm for text clustering. Their algorithm (due to its efficient and rapid response) applies to issues where data are not in uninterrupted spaces. It is also applicable to issues where asymmetrical affinities and the matches do not meet the triangle inequality. After evaluation, it was found that the default similarity measurement scheme (Euclidean) is not well-suited for clustering textual data. Several proximity measures were used; it was found that the Kullback–Leibler (KL) divergence is better than the Euclidean distance in this regard. Thus, we can conclude that to obtain better performance in text clustering, the KL divergence and the affinity propagation algorithm should be used jointly.

Dataflow clustering recognizes the radical dataflow patterns. The authors of [65] emphasized identifying the most relevant exemplars from the flow of data. Two issues must be considered: how to group by following the best exemplars and how to deal with the producing patterns, which are crucial properties of data streams with dynamic distributions. To meet the first challenge, the affinity propagation (AP) algorithm was employed. Then, change detection was applied to address the second problem. The recommended STRAP algorithm blends the AP to group data sets according to the best exemplar; meanwhile, the clustering model is recreated in case the evaluation identifies a change in the radical data distribution. Experimental results demonstrated that the recommended methods can accurately identify the best examples from the dataflow, and then build clusters according to these examples.

### 5.7. Utilities

Subbiah et al. [58] addressed the use of affinity analysis (AA) in a qualitative investigation to identify the critical factors for e-service challenges in Malaysia. They also demonstrated how affinity analysis can be employed to explain the relative data of an analytical method. The proximity analysis uses the NEWS matrix to provide a new experience in experimentally interpreting the quality of data; it was clearly shown that a similarity analysis of the flawless output can be used to mathematically evaluate qualitative data . This could help researchers to evaluate and perceive the level of confidence when identifying the root cause of the necessary critical factor. This analysis was applied to data from a pilot study consisting of focus groups and individual interviews.

The properties of related data groups regarding the familial faults of power transformers were observed and, in order to deal with these properties, several requirements were specified. To fulfill these requirements, Zhang et al. [71] introduced an adaptive repeated itemset mining algorithm (depending on the Apriori algorithm). Using this algorithm, the similarity analysis of familial defects of power transformers is executed in a vast number of records obtained from in-service power transformers. After evaluating a large data set, it is clear that the proposed algorithm can effectively specify the similarities among manufacturers, operating years, the voltage grades of transformers, and their defective parts.

Pitkaranta et al. [73] considered coded data sets, which are often used as brief representations of real-world procedures. A previously coded data set implies an approach that can only work with a small data set. Therefore, in this study, they introduced an information retrieval-based model that could work with a vast amount of data without facing any complexity. Their experiment used artificial coded data sets and data sets collected from independent domains. Real-world data collected from eminent domains (e.g., peddle businesses and healthcare) were used jointly as well as massive artificial coded data sets. By employing a kernel technique and vector distance functions, the proposed method indicates proficient closest neighbor queries. After evaluating the coded data set, the experimental results indicated that this model could perform effectively in modifiable queries along with massive coded data sets, generating rapid results.

**Table 3.** Summary of the research on affinity analysis.

| Paper | Major Contribution | Data Set Used | Implementation Environment and Evaluation | Limitations |
|---|---|---|---|---|
| Shyu et al. [45] | The authors proposed an algorithm for affinity-based association rule mining for discovering quasi-equivalence relations in database queries. | Experimented on the Purdue University financial database management system with 22 media objects accessed by 17,222 queries. | Conducted two empirical studies, which showed that their proposed approach performed better than the traditional association rule mining approach in discovering quasi-equivalence relationships. | As the structural similarity link between two media objects cannot be automatically discovered, the query execution speed of this approach is still slower. |
| Xiong et al. [56] | To capture the degree of affinity in a pattern and remove cross-support patterns, they created a metric called h-confidence and proposed an algorithm called the hyperclique miner. | They evaluated their proposed algorithm on the *Pumsb* data set containing 49,046 records, S&P 500 data set (716 records), and the *Retail* data set (57,671 records). | All tests were conducted on a Sun Ultra 10 workstation running the SunOS 5.7 operating system with a 440 MHz CPU and 128 Mbytes of memory. They performed better than existing algorithms used for identifying patterns at very low support levels. | In practice, cross-support patterns, which contain elements with varying amounts of support, are loosely related and are not suitable for real-world purposes. |
| Yun et al. [46,51] | A new algorithm called weighted interesting pattern mining (WIP) was proposed, in which a new metric—weight-confidence—was created to produce weighted hyperclique patterns of similar weight levels. | Three real data sets were used: *Connect, Pumsb*, and *Mushroom*. | The WIP algorithm was implemented in C++ and experiments were performed on a Unix platform. The algorithm was compared with WFIM [86] and hyperclique miner [56]. | This technique is centered on the support confidence structure. However, it does not take into account the actual preferences of users. |
| Changchien et al. [16] | The authors proposed an association rule mining mechanism based on the self-organization map (SOM), a neural network for affinity grouping, and then a rough set for the extraction of association rules. | Experimented on purchase records of an e-commerce store containing 1120 products and 35 customer records. They only considered records for one whole day of transactions, comprising 2000 records. | The radius and learning rate of SOM clustering were set to 2.0 and 1.0, respectively. The change rate for both radius and learning rate was set to 0.98. The error rate of the clustering was lower than 0.1. For rule extraction, the minimum confidence was set to 0.2. | This strategy cannot be used for dynamic assessment (i.e., it cannot automatically add feedback and attributes). One should consider more variables, such as customer and product profiles, purchasing purposes, and so on. |

**Table 3.** *Cont.*

| Paper | Major Contribution | Data Set Used | Implementation Environment and Evaluation | Limitations |
|---|---|---|---|---|
| Shyu et al. [62] | The authors proposed a new metric of similarity based on affinity to facilitate web document clustering. | *Microsoft Anonymous Web,* also known as the UCI KDD data set, contains 294 URLs and about 38,000 anonymous user accesses. | Four experiments were performed to validate the efficiency of the proposed affinity-based similarity measure. The experimental findings reveal that the suggested metric of the affinity-based similarity (AFFINITY) performs better in terms of generating the smallest number of inter-cluster accesses. | It requires a training data set, which intensely conditions the clustering results. This represents the main drawback of the approach. |
| Wang et al. [57] | The authors proposed a hierarchical technique, named weighted confidence (WC) clustering, for grouping market-basket data items based on a weighted confidence affinity measure and vertical data structures. | Standard market basket data sets and properties include transactions and itemsets. | The suggested weighted confidence affinity measure gives much more compact results than other concurrent affinity measures. Further, it determines the proximity between clusters more effectively than other affinity measurement approaches. | The PH-clustering scheme employs a larger number of processors to expedite the sequential clustering process. Due to the trade-off between computing time and communication time, it was noticed that expanding the amount of used processors does not really speed up clustering. |
| Chen et al. [63] | Addressed the problem of delayed diagnosis and, in order to decrease error probability, used the affinity set (through the topological concept) to classify/focus on key attributes that caused a delayed diagnosis (missed injury). | Clinical ER data of Kaohsiung Medical University and Chung-Ho Memorial Hospital, Taiwan. | Achieved accuracy of 89.4% when compared with the Rough Set, SVM, logistic regression, and neural network models. | Lack of implementation details and performance evaluation measures. |

**Table 3.** *Cont.*

| Paper | Major Contribution | Data Set Used | Implementation Environment and Evaluation | Limitations |
|---|---|---|---|---|
| Li et al. [69] | The authors proposed a group of content-abstracted items that induced a blogger's cascade joining behavior. | Data set consisting of blog posts published from June to September 2008. | It uses an SVM-based classification technique. Five properties—the number of friends, the number of candidates, the popularity of candidates, the time passed since the genesis of the cascade, and the citing factor of the blog—have notable impacts on the cascade affinity, helping to point out the bloggers who could easily influence. | As there are significant variations between retweeting in microblogs and blog cascades, the proposed solution employed in this paper may not be directly applicable in a microblogging environment. |
| Wang et al. [47] | The authors proposed a parallelized hierarchical clustering technique for market basket data and introduced a new WC affinity measure to enumerate the resemblance between clusters while reducing the impact of low-support items. | Market basket data properties include transactions and itemsets. | After they conducted an experiment on market basket data, it was evident that the recommended weighted confidence affinity measure produces much more compact results than several modern affinity measurement schemes. | This strategy cannot function in a dynamic context. It is performed on static data inputs and, so, is unable to generate real-time relationships and behavior patterns. |
| Panigrahy et al. [52] | The authors introduced a sketch-based new affinity measure in social networks that considers both the space between two individuals and total edge-separated paths between them. | The first experiment (Hotmail experiment) used three data sets: a) an anonymized data set with a pair of user ID hashes for any two email users, b) a data set consisting of basic demographic information, and c) a data set that includes the Bing query history. The second experiment evaluated one month's worth of Twitter posts. | In this sketch-based social affinity measure approach, affinity queries could be responded to much more effectively than others. | Given that the measure relies on metrics, e.g., the number of interactions between individuals, it may fail to capture the set of components that may better explain the affinity. |

**Table 3.** *Cont.*

| Paper | Major Contribution | Data Set Used | Implementation Environment and Evaluation | Limitations |
|---|---|---|---|---|
| Li et al. [48] | The authors proposed a group of content-abstracted properties (considering large-scale and small-scale) that induce a blog's tendency to enter into a cascade. | The data set contains blog posts published from June to September 2008. | Their approach identified a total of seven properties that played a significant role in predicting affinity in the blog cascade, pointing out the bloggers who could easily influence. | To adapt this model to the microblog domain, some parameters may need to be modified, and other characteristics specific to microblogs must be considered. |
| Li et al. [53] | The authors proposed a framework, *FRec*, recommending prominent users and topic-related interactive communities that are very compatible with the given user, depending on the presumed outcome of the topic model. | Data set was an accumulation of tweets linked with presidential campaigns between Barack Obama and Mitt Romney, ranging from March to May 2012. | Based on the input, the proposed framework, *FRec*, could suggest an array of prominent or interactive users that correlate with the context, and those most compatible with a given user or keyword. | It does not provide any viable strategy to filter out the noise within the social media data. |
| Rezgui et al. [49] | The authors introduced a system named *AffinityFinder* that utilizes sentiment analysis to avoid the complexity in finding potential hidden associations among Twitter users. | The data set included around 12,000 directed tweets by approximately 600 users. | The proposed system, *AffinityFinder*, is capable of deriving probable friendship connections with high accuracy and, according to non-traditional information, it can make a more relevant supposition about the association. | The proposed method is not a real-time system, the MongoDB database is not updated regularly with newer tweets, and the relationship scores are not refreshed periodically. |
| Hong et al. [54] | The authors proposed a novel recommendation technique to determine social affinity based on the closeness between two community members. | Synthetic data sets. | After evaluating the synthetic data sets, it was shown that the recommended scheme could efficiently discover social affinities. | The system lacks experiments with a broader consumer group (rather than 18 people). Additionally, a virtual network that mirrors the real world using a standard theory needs to be created. |

**Table 3.** *Cont.*

| Paper | Major Contribution | Data Set Used | Implementation Environment and Evaluation | Limitations |
|-------|-------------------|---------------|-------------------------------------------|-------------|
| Tshimula et al. [50] | The authors proposed a method to detect hidden affinity associations between individuals on the web. | Used four online discussion data sets: 1) WhatsApp group data (WGD); 2) R community data on Twitter; 3) FreeCodeCamp Gitter chat data (FCC); and 4) internet argument corpus. | HAR-search comprises much more properties than AffinityFinder and could notice every detail to observe the context thoroughly. Moreover, HAR-search produced more effective results compared to AffinityFinder on the experimental data sets. | HAR-search models only positive interaction sequence(s) (PIS) based on the text context in the conversation history. |
| Moraes et al. [60] | The authors introduced an algorithm to analyze the context-free grammar affinity using test sentences originating automatically and randomly. | The set of all sentences that could be obtained from grammar. | After the evaluation, it was shown that the recommended trial and error-based algorithm could effectively analyze context-free grammar affinity and produce the desired output. | This algorithm is not fully capable of being applied to practical systems as its computational cost is high. |
| Pitkaranta et al. [73] | The authors introduced an information retrieval-based model that employs kernel methods and vector distance functions for systematic analysis of coded data sets. | Artificial coded data sets to which some renowned domains were added, namely healthcare and market basket analyses. | This model can perform smoothly for modifiable queries with vast amounts of coded data sets, generating rapid results. | The proposed information retrieval model mainly focuses on achieving a high performance, rather than accuracy; hence, it does not deliver high accuracy. |
| Shen et al. [68] | The authors proposed a frequency-based similarity analysis to group arrays. An interdisciplinary analysis strategy was also utilized to reconstruct this array based on the access frequency. | Used 11 benchmarks: eight from SPEC CPU2000 and three from the distance-based affinity analysis programs by Zhong et al. [87]. | After the evaluation, the experimental results showed that the authentic compiler-based array rearrangement improved most of the program's performance compared to the others; no further improvement was required through code or data categorization. | Their static estimation only works for loops with regular structures and cannot accurately analyze irregular loops, indirect memory references, and complex dataflow with aliases. |
| Sadiq et al. [66] | The authors introduced a framework named the associative affinity factor analysis (AAFA) for position analysis and bot spotting; it recognizes actual users from bots and detects the positions within inconsistent affinities. | The 2016 U.S. presidential election campaign tweets were used as an experimental data set. | Their experimental results showed that the correctness of the suggested framework is very high when compared to various existing modern methods. | Other related information in tweets, such as resources, retweets, favorites, and so on, should be taken into account for enhanced stance recognition and bot categorization. |

**Table 3.** *Cont.*

| Paper | Major Contribution | Data Set Used | Implementation Environment and Evaluation | Limitations |
|---|---|---|---|---|
| Subbiah et al. [58] | Conducted a pilot study to determine the most relevant critical factor that led to dissatisfaction among the users in providing e-services. | A pilot study: 3 groups of 8 people were formed, for a total of 24 participants comprising residents from 13 states in Malaysia. | By using the affinity analysis, they determined the critical factors of qualitative data. According to the findings of the affinity analysis, 80% of the problem was caused by 20% of the overall functionality of the serving system. | This paper was a pilot study that involved only 24 participants; surveying with fewer data is more likely to produce misconceptions and erroneous results. |
| Zhang et al. [71] | The authors introduced an adaptive repeated itemset mining algorithm, which was improved based on the Apriori algorithm in order to obtain the precise requirements of mining fault information, thereby supporting an insightful condition evaluation system for power transformers. | A total of 11,266 authentic records from in-service power transformers were used in the experimental data set. | The recommended algorithm can effectively specify the similarities among manufacturers, operating years, the voltage grades of transformers, and their defective parts. | Lack of implementation details and performance evaluation measures. |
| Masum et al. [59] | The authors introduced a supervised learning technique to detect affinity based on assumptions acquired by tracing trajectories of the human context captured in a video series. | Used 30 video clips, each with a length of at least 6 seconds, as a data set. | After evaluation, this assumption-based similarity identification scheme showed optimal results regarding accuracy in similarity identification when compared with other similarity measurement schemes. | The proposed algorithm has high computational requirements. |
| Liu et al. [64] | The authors proposed an algorithm named *Affinity propagation (AP)* to cluster oral conversational text. | Used textual corpus as an experimental data set. | After performing affinity propagation on text clustering, it was shown that, if the Kullback–Leibler (KL) divergence and affinity propagation are jointly applied, a better outcome could be obtained, compared to when using the default measure (Euclidean). Furthermore, it can effectively handle noisy data. | This paper applies the AP–KL method only on a textual corpus; it needs to be investigated further in other data sets to demonstrate its performance. |

**Table 3.** *Cont.*

| Paper | Major Contribution | Data Set Used | Implementation Environment and Evaluation | Limitations |
|---|---|---|---|---|
| Zhang et al. [65] | The authors introduced the STRAP algorithm, blending affinity propagation (AP) with a statistical change point identification test to group data sets according to the best exemplars. | Two benchmark data sets. | The recommended method can accurately identify the best exemplars from the flow of data, and then build clusters according to these exemplars. | The proposed STRAP algorithms take a long time to compute. The question was raised as to whether or not this performance is compatible with real-time applications. |
| Srilatha et al. [67] | The authors proposed a unique high-utility pattern (HUP) discovery approach for progressive transactional databases by considering a resilient frequency affiliation (FA) and knowledge weight utilization, thus preserving the downward closure principle. | Three data samples—retail, food mart, and chess data information. | Experiments showed that the proposed itemset extraction method is excellent and scalable for mining intriguing HUP data. | The proposed data mining approach mainly focuses on improving the run-time and memory usage; other features, such as output accuracy and data security, were not considered |
| Swofford et al. [72] | A unique *Deep Affinity Network (DANTE)*-based approach for detecting conversational groups; it forecasts the affinity that two people in a situation are in the same conversational group based on the social environment. | This paper considers three publicly available data sets of social interactions for evaluation: the Cocktail Party data set [88], the SALSA data set [89], and the Coffee Break data set [90]. | In comparison with earlier work, this algorithm outperformed in conversational group identification, general group identification, and graph clustering tasks, when evaluated against the existing benchmarks. | The suggested technique ignores the temporal relationship of spatial data recorded by a sensor located anywhere. DANTE scales quadratically with the scene size; this could be a problem when used in a more crowded environment. |

**Table 3.** *Cont.*

| Paper | Major Contribution | Data Set Used | Implementation Environment and Evaluation | Limitations |
|---|---|---|---|---|
| Yadav et al. [75] | The authors proposed an HSIC-based affiliation approach for learning non-linear relationships between data points, which subsequently identifies and connects related data points. | Synthetic data sets comprising 3D manifold shapes with known lower-dimensional 2D representations: punctured sphere, sine on a hyperboloid, Swiss roll with hole, twin peak with hole. Real-world data sets: BCI 5F [91], BCI HaLT [91], Hasy_v2 [92], natural images [93], CVPR'09 [94], and fashion MNIST [95] | The presented HSIC-based affinity successfully binds identical data points together and separates different data points to give a more explicit portrayal of resemblance, compared to the Euclidean distance, and it yields high precision under several dimensionality reduction approaches, such as LE, LLE, and LTSA | Although HSIC may be a suitable affiliation indicator, the precision of classification relies on other factors, such as the number of neighbors in the graph, the choice of the classification scheme, and so on, which also need to be experimentally evaluated. |
| Bao et al. [76] | The authors introduced four new assessment measures (Bi-support, Bi-lift, Bi-improvement, and Bi-confidence) for mining association rules. | Two types of data sets were utilized in the evaluation: the IBM SPSS modeler data set, which has over 1000 shopping entries, including 11 items, and the groceries data set, which consists of a total of 9835 purchase records and 169 items. | The suggested Bi-support and Bi-confidence paradigms outperformed previous models, in terms of objective evaluation, extensive definition, and application in practice, with excellent precision and reliability for filtering and choosing frequent patterns. | They did not test for large amounts of data (e.g., zettabyte-scale). The second issue is that experimental results in different relevant domains were inadequate. |

The study in [68] presented an affinity model based on frequency, context-sensitive interdisciplinary evaluation, fixed approximation, and lightweight profiling of the implemented frequency. The authors also showed that scientific programs do not require the overhead needed to rearrange arrays, such that we can obtain the same benefit through a precise program analysis. The analysis schemes were applied in producing the basic structure of the compiler and were examined in SPEC CPU2000 FORTRAN benchmark programs. After the evaluation, the experimental results showed that novel compiler-based array rearrangements improved most of the program's performance, compared to others, and no further improvement was required through code or data categorization.

## 6. Review of Bibliomining

A significant number of bibliomining methods were recently reported on in the literature (Table 4). Azam et al. in [25] discussed the different stages and procedures used for mining information from the North South University (NSU) library and concentrated on valuable examples among borrowers utilizing bibliomining methods. The findings of this process may be utilized to achieve more proficient administration and spending schemes for the library. Certain methods (e.g., grouping) could be investigated to better understand the conduct of library clients.

Nicholson et al. in [28] considered a canny specialist for the advancement of an automated collection in a computerized library setting. They utilized a predictive model dependent on aspects of every web page to identify insightful works. The subsequent models could be utilized in the choice cycle in order to construct an advanced library of web-based academic exploration works. The procedure can be extended to obtain a computerized library of organized electronic data.

The work by Nicholson et al. [21] considered the combination of data mining techniques in computerized library administrations. Bibliomining or a combination of bibliometric and data mining methods were characterized, in terms of comprehending library administrations. A preferable system for bibliomining from the perspective of the library chief and the library specialist were introduced and analyzed. Finally, an examination plan for determining a large number of fundamental bibliomining issues and to push the field ahead carefully was created. The outcome is a guide used to consider the combination of data mining in advanced library administrations and it provides a format that other cross-discipline data mining analysts can follow for methodical investigations in their respective subject spaces.

Nicholson et al. in [26] presented a global perspective on the information produced in libraries and the variety of ways in which library and data administrators can utilize information mining in their libraries (e.g., bibliomining) in order to comprehend examples of conduct among library clients and staff, as well as examples of data asset usage throughout the foundation. They looked at information sources and the potential utilization of information mining strategies, and investigated the legal and moral ramifications of data mining in libraries. Another work, [78], deconstructed evidence-based Librarianship to show the clients that were included, depending on the information assembled in libraries. The bibliomining cycle, or a combination of information warehousing, data mining, and bibliometrics, is utilized as a system to construct an alternate means for EBL. Bibliomining—and evidence-based Librarianship—is not proper for all subjects; if the curios of the library can be assembled and investigated, this strategy could provide an alternate means to arrive at the EBL objectives.

The objective of [22] was to clarify the bibliomining procedure. Priority was placed on information warehousing and benefactor security issues (as they are needed prior to whatever else is needed). It is fundamental to catch our information-based institutional documents while ensuring the protection of clients. By utilizing an information stockroom, these two objectives can be met. When the information stockroom is set up, the library can utilize investigation devices to obtain more exhaustive information on the client networks and asset usage.

Hajek et al. in [70] proposed the use of clustering methods to aid libraries in deciding the degree of public administration, by deciding commonplace customers in a manner not founded on financial allotment or on the assessment of public administration, in order to acquire standards of conduct for delegate clients. The consequences of this work could then be executed to accomplish more productive administration of public libraries. This method constructs a strategy for bibliomining, including the qualities of the cycle's different phases. The outcomes are explained; a proposition for continuing an examination is also included.

Utilizing clustering methods and techniques to extract qualities from genuine public library information to reveal the likenesses in administrations given by open libraries, in light of personal conduct standards of the clients, was discussed in [23]. The outcomes (from utilizing measurable techniques) demonstrated that the essential, optional, and tertiary library administrations are given in combination, and that they supplement each other; there is a reliance on the necessities of clients accomplishing the more productive administrations of public libraries.

Shieh et al. [24] proposed a novel bibliomining model and constructed a bibliomining incorporation model framework to test the practicality of the proposed model. The essential purpose of bibliomining is to find the essential and valuable data that help in the decision making of library directors, who should give much consideration to the most proficient method to meet their necessities. The created bibliomining incorporation framework meets this purpose and can proficiently assist librarians in carrying out data mining functions.

Deshpande et al. in [27] analyzed knowledge on the utilization of library books among undergraduate computer science students through a library book data set obtained from a computerized library. They found that the students made ideal uses of the library books; they additionally found that female students used the library more than males. The library utilization was lower in 2011, although there was an expansion in library use again in 2012. The examination was carried out utilizing the XLminer data mining instrument.

Prakash et al., in [77] reviewed the data sources and potential uses of data mining procedures in libraries. The way toward utilizing library information more successfully begins with finding approaches to associate the unique wellsprings of information that most libraries provide. Associating these different sources in information distribution centers can encourage methodical investigations with various devices, in order to determine the standards of conduct of a library's essential body electorate. These examples can assist in improving the library experiences of clients, assist library executives in making choices and setting strategies, and help parent associations or networks to comprehend the data required of their members.

The motivation behind the research in [80] was to narrate a novel methodology to recognize the pathways through which examinations may affect other explorations, innovation advancements, and functions, distinguishing the specialized and framework qualities of the client populace. A novel writing-based methodology was created to distinguish the client network and its attributes.

Kao et al. in [82] considered the significance of flow information preparation in detail; they presented a spending allotment model by utilizing an information mining method, delineated the utilization of ABAMDM, and constructed an application. The significant concern regarding this study is that the spending designation should have the option to mirror a pre-requisite: the more an office utilizes its gained information in the current scholastic year, the better the financial plan for the upcoming year. The essential yield of ABAMDM, used to infer loads of obtaining spending distribution, comprises two sections: one involves expressive information through usage focus and the other is the reasonableness employing association for the concerned divisions.

**Table 4.** Summary of the research on bibliomining.

| Paper | Major Contribution | Data Set Used | Implementation Environment and Evaluation | Limitations |
|---|---|---|---|---|
| Azam et al. [25] | The authors proposed different stages and techniques for mining the North South University (NSU) library information, concentrating on helpful examples. | Experimented with a database based on MySQL to determine the relationships between book classes. | Analyzed the NSU library database, successfully using bibliomining techniques (with 97% accuracy). | This article discusses different techniques for mining data and extracting patterns with respect to only one private university (North South University) library in Bangladesh. The system lacks application in a large domain. |
| Nicholson et al. [28] | Various bibliomining procedures were used to decide the best arrangement of rules to discriminate the different works; the best model delivered by every strategy was tested using an alternative arrangement of web pages. | Different data criteria were used for web pages. Then, a model was created, the data were queried and cleaned manually, and a data set was prepared for development and testing. The final data set was composed of 1000 documents that were selected for testing. | Used different models, such as classification tree, discriminant analysis, neural network, and logistic regression, for classification. Every model had a different accuracy and return. | The data used in this method were not real web data, and the amount of data were small. The precision of existing models could be improved by considering more real web data and applying feature reduction techniques. |
| Nicholson et al. [21,22] | The origin of the term bibliomining was investigated; the authors explored the link between bibliomining and its two key components: bibliometrics and data mining. A conceptual framework for library and information scientists was proposed. | Different analysis tools and data mining algorithms used in the data warehouse. | In two contexts—digital library administration and digital library research—the conceptual positioning of bibliomining (with respect to other types of evaluation) was discussed. Finally, a research plan was devised to address frequent concerns and to support the systematic expansion of bibliomining. | Data mining and bibliometric techniques were integrated to facilitate digital library services, but the implementation of these techniques was absent. |
| Nicholson et al. [26] | The authors investigated how data mining may be used by libraries and information management to better understand the behavioral patterns among library users and staff, as well as the design of information resources used across an institution. | Available data sources, such as integrated library system (ILS) data sources and external data sources, were used. | Data sources and potential data mining applications were investigated, as well as the legal and ethical aspects of bibliomining. | No mention about the utilization of library information to help vital administration choices for libraries and their host establishments. |

**Table 4.** *Cont.*

| Paper | Major Contribution | Data Set Used | Implementation Environment and Evaluation | Limitations |
|---|---|---|---|---|
| Nicholson et al. [78] | Presented a new approach to evidence-based librarianship (EBL) concepts. | Used multi-library data warehouses for traditional evidence-based librarianship. | For the measurement and evaluation of library services, they presented a different approach for combining data mining and data warehousing with bibliometrics. Standards for data collections and associated tools different from traditional EBL were proposed. | The main challenge of this article was to allow librarians to carve out the opportunities to gather information. |
| Hajek et al. [70] | Built a model of bibliomining considering the characteristics of the different stages of the process. | Used empirical data to determine a strategy to scrutinize the respondents regarding their singular preferences of public administrations given by the library, considering the inquiries for the valuation of their significance. | The $k$-means clustering algorithm was used to identify the services of public libraries. The clustering accuracy was higher (37% of users in total) when compared to other clustering approaches (10%). | A strategy was built for bibliomining, but the technique has numerous obstructions to its utilization. One significant drawback is the lack of a comprehensive analysis of the services provided by public libraries. |
| Hajek et al. [23] | The authors proposed a function for the grouping of strategies and extraction of characteristics from genuine public library information. | Used empirical data to check strategies for scrutinizing respondents based on their singular preferences of public administration given by the library, taking into account inquiries to account their significance. | For the identification of library services, data were collected, pre-processed, extracted, and then subjected to clustering methods. The results of ten components contributed to 54.45% of the variance of the main data. The results were obtained by using Ward's algorithm and statistical methods, according to the needs of the users. | The proposed method was not created exhaustively in the area of public administration, even though it is expected to help in the main itemized examination of the arrangement and provide information for streamlining executive processes in the association. |

**Table 4.** *Cont.*

| Paper | Major Contribution | Data Set Used | Implementation Environment and Evaluation | Limitations |
|---|---|---|---|---|
| Shieh et al. [24] | The authors proposed a bibliomining model that constructs an integrated arrangement for librarian bibliomining, ensuring the versatile and straightforward operations of library data mining activities. | Various data sources, including different databases and data files, were used. The data sources can include MS Access, MS Excel, SQL Server, and text. | Based on the bibliomining application model, a prototype for the integration system was developed in order to test and verify its feasibility. | Used a rapid prototyping development procedure to construct an integration system for librarians, which has lower accuracy and is more expensive. |
| Deshpande et al. [27] | Analyzed the usage of library books among undergraduate computer science students using a library book data set captured from an automated library | Considered 482 volumes of books from the library database between 2010 and 2013. XLminer, a data mining tool, was used for data analysis | Using the bibliomining technique in a year-wise manner, the library books among undergraduate computer science students were considered, such that the students could optimally utilize the library books. | Library use was lower in the year 2011, even though there was an expansion in library utilization in 2012. |
| Prakash et al. [77] | Provided an overview of data sourcing and essential functions of the data mining process in a library. | Used library data sources. | Data mining approaches work with classes, clusters, patterns, associations, and sequences. Different levels of analyses, such as ANN, DT, data visualization, genetic algorithms, rule induction, and *k*-nearest neighbor methods are available. | An outline of information sources and potential utilization of information mining in the library is presented, but any implementation is absent. |
| Kostoff et al. [80] | The authors proposed a novel approach for recognizing the pathways through which exploration may affect research and technology development as well as to recognize specialized framework attributes. | Selected 307 articles accessed in the Science Citation Index (SCI) database. | Applied text mining for the taxonomy of citing articles. Bibliometrics was applied to profile user characteristics. Different types of cluster analysis techniques were performed on the information extracted from the 307 first-generation cited articles. | Did not consider second- or higher-order citations. Moreover, the proposed approach was only applied to 307 fundamental physics articles. More domains and articles needed to be considered for better understanding and applicability. |

**Table 4.** *Cont.*

| Paper | Major Contribution | Data Set Used | Implementation Environment and Evaluation | Limitations |
|---|---|---|---|---|
| Kao et al. [82] | Presented a methodology (ABAMDM; acquisition budget allocation model via data mining) that specifically addressed the use of descriptive information acquired in historical circulation data to aid in library acquisition budget allocation. | Used a circulation database with various attributes, in which the circulation data were first pre-processed and in which performance and descriptive knowledge could be derived. | The proposed library budget assignment arrangement model could be used by an originator in developing a choice decision support network. ABAMDM and ARPM were applied in 17 departments and successfully found the number of records and final budgets of these 17 departments. | Information assortment through daily dissemination work is incredibly impacted by how a client utilizes web-based materials, thus making the budget plan portion of the activity considerably more troublesome. |
| Wu et al. [83] | A data mining model named DMBA was proposed to assist in apportioning library information to obtain a spending plan. | Used a data table that incorporated departments, materials used, and etymological strength (regarding the extent that a used material was identified with a department). | Developed an algorithm consisting of SQL statements. The outcomes with ARPM, DeC(Dpt), UGain(Dpt), BAWeight(Dpt), and the last library assigned procurement financial plan for divisions exhibited the utilization of DMBA and the outlined function for LKSUT. | The proposed budget plan assignment is complex, troublesome, and based on historical data. Their budget assignment methodology is outdated with respect to modern library requirements. |
| Butt et al. [55] | This article focused on a bibliometric examination and correlation of all distributed archives from 2015 to 2019, considering diaries in the review subject class of 'Statistics and Probability' under the classifications of Q4-IF and the Emerging Source Citation Index (ESCI) of the Web of Science (WoS). | All published documents from journals in the study topic category of 'Statistics and Probability' for Q4 Impact Factor (IF) journals and Web of Science's Emerging Source Citation Index (ESCI) journals from 2015 to 2019. | The greater part of the distributed records were multi-creator archives, yet a somewhat larger number of single-wrote records (23% and 22%, respectively) was noticed in both Q4-IF and ESCI journals. | This article used limited data from the literature; the time frame was also limited. |

**Table 4.** *Cont.*

| Paper | Major Contribution | Data Set Used | Implementation Environment and Evaluation | Limitations |
|---|---|---|---|---|
| Wen et al. [79] | Mapping of the knowledge domain was employed as an analytical approach to conduct a systematic analysis of the literature concerning building information modeling (BIM) research from 2010 to 2019. | A total of 1369 relevant published studies on BIM in the core Web of Science database. | For the analysis of data, the proposed method employed cluster analysis, co-citation analysis, and co-word analysis. The mapping of the knowledge domain was performed using CiteSpace software. The findings demonstrate that BIM research is still in its early stages. In addition, BIM research is primarily spread over nine domains. | They worked with a few specific fields; in their study, there was difficulty in summing up the turn-of-events pattern of BIM, which was obtainable from the small review test. |

Wu et al. in [83] considered the estimation of dissemination information preparation in detail, built a decision-making model named DMBA, which integrates the information mining strategy with the library spending portion activity, exhibited the utilization of DMBA, and outlined the LKSUT function. The created model depends on the ID3 calculation to investigate the illustrative information through data hypotheses and insights, in order to infer the suitability of a decision. The primary yield of the DMBA involves the library information premises, which could assist in defining the spending distributions for offices by utilizing a combination of the investigated logical information and its relevance. A reasonable application case evidenced the practicality of the created DMBA.

But et al. in [55] focused on the term 'statistics and probability' over the course of the past few years. Statistics and probability have become indispensable subjects that provide ways to manage designs and to obtain knowledge through information. Bibliomining and bibliometrics are reciprocally utilized, giving an entryway to assess such procedures and fill in the information gaps.

The study in [79] employed a co-citation analysis, co-word analysis, and cluster analysis to examine data, and created mapping knowledge domains using the CiteSpace software based on 1369 relevant articles in the published literature and the core Web of Science database. Their outcomes showed that building information modeling (BIM) research is in a rapid development stage. The exploration areas of interest are fundamentally moving in eleven directions, which can further be grouped into three levels. This paper presents essential hypothetical and common-sense references for the future examination of BIM.

## 7. Review of Technology Mining

Not much research has been conducted in the field of technology mining; however, some technological mining methods have recently been reported on in the literature (Table 5). Madani et al. [31] illustrated the development of technology mining research and how technology mining has quickly developed in innovative executive research. The bibliometrics instrument CiteSpace was utilized to handle the information of papers, combining the titles, abstracts, keywords, and citations to examine the references and substances in analytical writing. South Korean researchers, particularly four analysts who had the greatest effect on the technology mining region among technology mining creators, are highlighted in this research. More technology mining-related papers have been distributed in journals; 'scientometrics' and 'innovative forecasting and social change' identified with innovation. Moreover, the top five journals contained more than 50% of papers in the technology mining field. Since 2010, around 70% of technology mining papers have been distributed. From this examination, it is clear that the technology mining field has logically expanded in recent years.

In [30], cutting-edge small firms were considered to be adaptable and inventive. Productive, independent ventures effectively utilize outer data. These associations are forced to stay creative in severe and relentless climates. They face limitations due to expenses related to the advancements of constant development. Zones for development and proposals to address those territories were additionally introduced for future surveys and research.

Li et al., in [61] presented data and described how to develop a fine-grained innovation organization. They utilized the SAO structures extracted from patent archives in order to group specialized data and examine the semantic relationships among the specialized data. They proposed the construction of single- and multi-type innovation organizations, which could provide various specialized data and offer solid help for innovation advancement. Nano-compost licenses were chosen for contextual analyses, in order to show how to develop a fine-grained innovation organization. First, patent information should be gathered, including unstructured information and organized information from patent information databases. Then, through common language handling, SAO structures can be extracted from patent reports. From this point forward, tech watchwords should be recognized and the pairwise connections among tech catchphrases should be measured.

The technique proposed by Yoon et al. in [84] is an elective strategy for a patent reference investigation. A network investigation indicates the relationships among licenses as a form of visual organization, helping the analyst to naturally fathom the general structure of a patent information base. Second, the network investigation enhances the expected utility of the patent examination, as it considers more catchphrases and can deliver more essential markers. Third, the proposed strategy is more efficient, regarding search time and cost, as it changes unique reports into organized information through a content mining procedure. As a basic assertion, the overall foundation of organization examination and text mining was introduced in this exploration, portraying a general cycle of creating a patent organization system and directing the patent investigation. To this end, the development of such new lists as the innovation centrality file, innovation cycle file, and innovation watchword bunches was recommended. A commendable case is utilized to demonstrate the cycle of examination and to guarantee the utility of utilization. Finally, the ramifications of subsiding, the flow examination, and future exploration issues are discussed.

In [74], the authors studied how Stack Overflow clients label their inquiries, i.e., with the fundamental innovation of the inquiry cycle and the proposed affiliation rule mining and network identification methods to mine innovation scenes from the Stack Overflow question labels. The mined technology associative network (TAN) captures a broad scope of advances, the perplexing connections among the advances, and the pattern of advances in the conversations of designers on Stack Overflow. The creators constructed a site (https://graphofknowledge.appspot.com/, accessed on 7 February 2022) for the network to assess the mined innovation scene, the visit measurements of which (by Google Analytics) indicate the overall advantages of the innovation administration scheme for engineers. The site's utilization information, though restricted, gives initial proof of the interest in and value of the mined innovation scene.

Porter et al., in [29] presented a method to productively create compelling knowledge on new innovations, drawing on observations and bibliometrics to mine the data accessible in significant public electronic information bases. It utilizes new programming to speed up the optional examination of the information-based subject of interest searches. Likewise, the scopes of conceivable data profiles are delineated by inspecting innovative work (R&D) distributions and licenses relating to hardware gathering.

A reasonable model [85] for understanding innovation development, which features dynamic and profoundly associated connections among numerous advances, was proposed. An examination showed that, rather than thinking about advancements in disconnection, innovation development is best seen as a robust framework or a biological system that incorporates an assortment of inter-related advances. In particular, three jobs were identified that advanced play inside an innovation environment: parts, items/applications, and backing and framework. Advances in biological systems are associated with these jobs, affecting the development of each other. Grouped associations between innovation jobs are considered a means of impact. An experiment considering the utilization of the proposed model through instances of remote systems administration (WiFi) advancements, and a business smaller than the typical case in the computerized music industry, was carried out.

Zeba et al. [81] offered a visual and extensive investigation of the utilization of man-made reasoning in assembling. Existing insightful research on computerized reasoning in assembling (in the Trap of Science Center Assortment data sets) was inspected for the periods 1979–2010 and 2011–2019. These periods were considered separately when developing the term business 4.0. bibliometric, when the content examination of applicable writing was directed, and critical discoveries were recognized in this way. The outcomes indicated that the main themes at present are actual digital frameworks and brilliant assembling, profound learning and massive information, and continuous booking calculations. This exploration incorporated a detailed audit of important logical writing, utilizing bibliometric and content examinations to distinguish research trends, as well as provide examples of research points to artificial intelligence in assembling.

**Table 5.** Summary of the research on technology mining.

| Paper | Major Contribution | Data Set Used | Implementation Environment and Evaluation | Limitations |
|---|---|---|---|---|
| Madani et al. [31] | Provided an in-depth analysis for detecting and visualizing emerging trends in technology mining. | Applied appropriate queries on the Web of Science database; 143 papers were selected for analyses. | Applied a Java application named CiteSpace to sort out, recognize, and visualize patterns for envisioning, breaking down references, and evaluating substances in analytical writing. The analysis was divided into eight different clusters, based on five aspects. | The cluster analysis was performed based on only five aspects. More important aspects, such as evaluation and accuracy, need to be considered. |
| Trumbach et al. [30] | The authors proposed technology mining as a means of assisting smaller firms in staying informed about new ideas. | Tested the data regarding some typical issues of a fleet management company (FMC), comprising common zones for development in small innovative firms. | Identified the problems in high-tech small firms and developed recommendations that should benefit FMCs or any small business. The recommendations focused on how technology mining strategies could improve some of the recognized problems. | No mention of the scalability of the work for a larger company than a FMC. |
| Li et al. [61] | The authors proposed building a fine-grained technological network that could aid in identifying fundamental and emergent innovations in the nano-fertilizer field. | Utilized a patent database to gather patent information, including unstructured information and organized information. Nano-fertilizer licenses were chosen as part of a case study. | Technical information using subject–action–object (SAO) structures taken from patent filings were categorized; the semantic links between different forms of technical information were analyzed. Finally, a fine-grained method for the construction of a technology network was proposed. | The input data analyzed for the study were not sufficient. |
| Yoon et al. [84] | The authors proposed a network-based patent examination method, which is an elective strategy for a reference investigation. | Used networks of WDM-related patent documents mined from the U.S. Patent and Trademark Office (USPTO: www.uspto.gov) database. | Actualized an organization-based patent investigation strategy; the initial steps involved data assortment and data pre-processing. The patent organization was produced with hubs and connections. Finally, the patent investigation was completed. | It can be difficult to generate a patent community if the dimensions of the patent files are too large. The improvements of different quantifiable indices are required to amplify and spread the evaluations. |

**Table 5.** *Cont.*

| Paper | Major Contribution | Data Set Used | Implementation Environment and Evaluation | Limitations |
|---|---|---|---|---|
| Chen et al. [74] | Suggested applying affiliation rule mining and network recognition strategies to mine the innovation scene from the Stack Overflow question labels, where the mined innovation scene was referred to as a graphical technology associative network (TAN). | Utilized the Stack Overflow information dump delivered in March 2015, with information from 2008–07–31 to 2015–03–08, containing 7.89 million inquiries that were joined with at least two labels, and 39,948 remarkable labels from these inquiries. | The study showed that the mined TAN caught a wide scope of innovations, the intricate connections among the advances, and the pattern of the advances in the conversations of engineers on the Stack Overflow. | To enhance the website design, it was necessary to gather extra fine-grained consumer interplay information, provided along with a cursor hover and a right-click inside the TAN. |
| Porter et al. [29] | The authors proposed a semi-automatic trend detection system named technology opportunities analysis (TOA) for the analysis of technological opportunities relating to emerging technologies. | INSPEC was used as the database for the sample searches, which is a very rich abstract database. | Documents were extracted from the INSPEC database using keywords. Then, the formed queries were provided to custom software, Technology Opportunities Analysis Knowbot (TOAK). Finally, the bibliometric analysis was performed for TOA. | TOAK is designed to be utilized interactively and iteratively by a human expert, which makes it unsuitable for general use. |
| Adomavicius et al. [85] | The authors introduced a model to understand innovation development through the perspective of an innovative biological system. | Considered technology classes, such as desktop PCs (Dell XPS 600 gaming PC), wireless networking (Wi-Fi) technologies, and a business mini-case on the digital music industry. | Utilized various levels of ideas to distinguish the structures of related innovations. The perspectives on biological systems of PC advances incorporate PCs as contending innovations and supporting advances. The advancements were quickly distinguished, identified through the central innovation inside a given setting, dependent on the three jobs that advance play in an environment. | It could be fruitful to broaden a few new modeling factors in the framework in order to symbolize era function shifts over time. |
| Zeba et al. [81] | Incorporated a precise audit of logical writing utilizing bibliometric and content examination in an attempt to distinguish research trends and give examples of research points on artificial intelligence in the assembling field. | Bibliometric examinations of the obtained articles were first performed for the periods 1975–2010 and 2011–2019. | For the two time frames, they performed bibliometric and content investigations. For content investigations, they utilized WordStat 8 programming. | They only utilized the terms 'producing' and 'fake knowledge' in the inquiry. |

## 8. Discussion

Prior research on affinity analysis appears to have focused on the specializations of fundamental association mining algorithms, many of which are still in the early stages of development. These disciplines include measures of interest, domain knowledge and semantics, quantitative mining, disassociation mining, privacy mining, incremental mining, iterative and interactive (or guided) mining, and higher-order mining. Data mining techniques have been used in libraries to better understand the patterns of behavior among library customers and staff, as well as the patterns of information resource utilization throughout libraries. Despite the vast quantity of studies conducted on big data, few have considered how they might be used in academic libraries, including the methodologies and tools used to analyze the data. We conducted this review to find appropriate answers to our research questions from the selected primary studies. For a better understanding of the research in affinity analysis, bibliomining, and technology mining fields, the associated challenges, open issues, and future research directions are detailed in this section.

### 8.1. Answers to the Research Questions

This section provides the answers to our research questions:

**RQ1: what are the different types of application domains of affinity analysis?**

Association rules are now widely employed in various disciplines, such as prediction, tailored recommendations, risk analyses, and others. Initial research focused on the application of association mining in market basket analysis [16,47,57]. In our review study, we found various application domains of affinity analysis, as described in Section 5. These domains include social network analysis (Section 5.2), natural language processing (Section 5.3), video analysis (Section 5.4), healthcare (Section 5.5), affinity propagation (Section 5.6), and utilities (Section 5.7).

**RQ2: how are affinity measures incorporated to improve itemset mining?**

The problem of discovering correlations between objects in a data set—known as association (rule) mining—has received much interest, especially after the publication of the Apriori algorithm. Market basket data, which enable corporations to better understand purchasing behavior and, as a result, better target market audiences, were a major driving force behind the initial research. Various affinity measures have been incorporated in association mining for better performance. Furthermore, various approaches have been proposed in the literature to improve the overall operation of association mining [32,76,96]. A new combined support value was introduced in [97]; it is used in combination with the Apriori algorithm to create package elements that are within a minimum support value. A novel bidirectional affinity measure framework was proposed in [76] to improve the existing association mining approaches. A novel algorithm for affinity-based association rule mining was proposed in [45]. In [16], the authors introduced a new metric called h-confidence. Wang et al. [47,57] introduced a new weighted confidence (WC) affinity measure to enumerate the resemblance between clusters while reducing the impact of low-support items. The usual approaches for evaluating association rules, which use support, confidence, and accuracy, have several drawbacks. Improvement, conviction validity, lift, Chi-square analysis, and other new evaluation approaches have been discussed in several studies.

**RQ3: what are the key bibliomining practices and technologies available for academic libraries?**

The key bibliomining practices we observed in our literature survey are listed below:

- The term "bibliomining" was first used by Nicholson et al. [22] to represent data mining for libraries. They presented bibliomining as a combination of bibliometrics, data warehousing, and data mining [21].
- Kao et al. [82] utilized bibliomining as an essential tool to uncover useful library information from historical data to aid decision-making; however, a systematic pro-

cedure was missing for appropriate knowledge discovery. Bibliomining was used in conjunction with other measurement and assessment methods to generate complete reports on library systems, in [28,78].

- Different types of data mining approaches, such as association, classification, regression, clustering, and neural network methods have been applied for the analysis of library data. An association rule-based method was proposed by Wu et al. [98] to evaluate the usefulness of a library collection.
- Decision/classification-tree based approaches were proposed by Kao et al. [82], Nicholson et al. [22], and Wu et al. [83].
- Clustering-based approaches for bibliomining analyses were proposed by Hajek et al. [70] and Kostoff et al. [80].
- Bibliometric analyses for university libraries were performed by Azam et al. [25] and Deshpande et al. [27].

A thorough review of bibliomining can be found in Section 6. A summary of the associated studies, along with their contributions, used data sets, evaluations, and limitations, is presented in Table 4.

**RQ4: what are the existing techniques and approaches for technology mining?**

The major existing techniques for technology mining, as determined from our selected literature, are listed below:

- One of the prime fields of technology mining is bibliometric analysis or text mining. Several approaches were found for technology-based text mining. Madani et al. [31] proposed a method based on articles found in the Web of Science database. Another approach based on the Web of Science core collection database was proposed by Zeba et al. [81].
- Another core field of technology mining involves patent analysis or patent mining. Network-based patent mining was proposed by Li et al. [61]. Another approach considering the U.S. Patent and Trademark Office database was proposed by Yoon et al. [84].
- Innovation mining is another promising aspect of technology mining. Porter et al. [29] proposed a method—the technology opportunities analysis, TOA—for innovation mining. Further, a model developed from the perspective of an innovative biological system was described by Adomavicius et al. [85].

A detailed review regarding technology mining is provided in Section 7. The summary of the associated studies, the contributions, used data sets, evaluations, and limitations are presented in Table 5.

**RQ5: what are the existing open issues, key challenges, and future research directions in the fields of affinity analysis, bibliomining, and technology mining?**

The following Sections 8.2 and 8.3 describe the answer to this research question.

*8.2. Challenges*

In every research work, there are always challenges. In our literature study, we found several challenges relating to different domains. Here, we mention some of the challenges that are most common in the domains of affinity analysis, bibliomining, and technology mining:

**Scalability:** Scalability refers to a system's capacity of adapting to new demands at a larger scale in the future. According to the reviewed literature, many of the proposed methods for affinity analysis, bibliomining, and technology mining systems operate on limited scales, and the validity of these approaches is only guaranteed within a small range. Scalability is a crucial consideration; however, the offered approaches were employed mainly in constrained settings.

**Mobility:** Mobility refers to the ability of a system or a model to be accessed from anywhere at any time. Due to the rapid growth of information technology, mobility is a significant factor that has received little attention in the literature.

**Systematic Comparison:** The majority of the literature reported on model performances based solely on the achieved results with a few relevant metrics, rather than a model's efficiency combined with quality metrics. Most of the studies lacked systematic comparisons, making it difficult to determine the applicability of any system in that domain.

*8.3. Open Issues and Future Research Directions*

- The need for a systematic comparison method for an affinity-based model is still a major open issue. Research is required for the development of appropriate comparison criteria.
- An affinity analysis in social media is an emerging and very crucial application domain [49,52,53]. Most research studies have focused on the affinity within a single social media platform. This research can be extended to incorporate the affinity measure of a user (or group) between multiple social media platforms.
- Applying affinity analysis methodologies in developing fields of study (e.g., sentiment analysis, text classification, Internet of Things, and sensor networks) could open up new possibilities.
- Data privacy and security have always been prime concerns in data mining approaches. This is a crucial issue, especially in bibliomining and technology mining. Various types of data are associated with academic libraries, such as user credentials, digital article copies, billing information, and subscription data. The privacy and security of these data need to be guaranteed. Few approaches regarding privacy and security in bibliomining can be found in the literature. Securing data privacy can be considered a potentially fruitful research direction.
- The application of blockchains in various aspects of technology mining is still in the emerging phase. Blockchain is a cryptography-based decentralized database system, which can be very handy for securely storing associated data and managing authentication schemes.

## 9. Conclusions

In this literature review, we aimed to make it easier for interested readers and practitioners to learn about research in the affinity analysis, bibliomining, and technology mining fields. To accomplish our goal, we conducted a detailed analysis of the literature focused on the use of data mining techniques in the mentioned fields, with an emphasis on case studies published between 2000 and December 2021. We performed this review to satisfy our pre-defined research questions. From 1073 initially selected papers, after applying various inclusion and exclusion criteria, we finally selected 53 primary studies for our review. We presented a comprehensive review, highlighting the significant contributions, used data sets, implementations, evaluations, and limitations of the selected studies. Our findings indicate that the application of an affinity analysis is not confined to association mining and market basket analysis. We described recent research approaches to affinity analysis in various application domains. We also considered the bibliomining process, which combines bibliometrics, data warehousing, and data mining to discover patterns, and we explored the concept of bibliomining with the integration of data mining in digital library services. We observed that the primary motive of bibliomining is to find significant and helpful data to aid in the decision-making of library directors. Existing technology mining techniques were also discussed. The main limitation of our review is that there were no standard performance-measuring criteria available to evaluate the works in our three different fields. We pointed out the challenges that are common in the fields of affinity analysis, bibliomining, and technology mining. Finally, some open issues and future research directions were presented. We believe that this study could be a valuable 'staging point' to further explore the fields of affinity analysis, bibliomining, and technology mining.

**Author Contributions:** All authors contributed equally to this work. All authors have read and agreed to the published version of the manuscript.

**Funding:** This research was funded by the Institute of Information and Communications Technology Planning and Evaluation (IITP), grant number No. 2020-0-00347.

**Acknowledgments:** This work was supported by the Institute for Information and Communications Technology Promotion (IITP) grant funded by the Korea government (MSIT) (No.2020-0-00347, Development of JPEG Systems standard for snack culture contents).

**Conflicts of Interest:** The authors declare no conflict of interest.

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
