# Peer review of "A Comprehensive Survey on Affinity Analysis, Bibliomining, and Technology Mining: Past, Present, and Future Research"

_applsci, doi:10.3390/app12105227_

Round 1

Reviewer 1 Report

The topic of the research proposed by the authors is current.

Please find my detailed comments below:

The authors must follow the instructions for the research manuscript sections. The paper must contain the required sections. The paper should have a clear structure, including an introduction, materials and methods, results, discussion, conclusions (optional).

Abstract

It should contain a synthesis of the research, methodology, obtained results and conclusion. In this case, it is a description of the current context.

Introduction

Regarding the introduction, I would recommend some more explicit answers to the following questions:

  1. why is the paper interesting?
  2. what is the purpose of it?
  3. what are the research hypotheses?
  4. what are the main results?
  5. who are the users of your research results? How can this results be used for scientific purposes?

It is hard to convince the reader about the importance of the paper without having immediate answers to the questions mentioned above. 

Results and Discussion

This section does not exist.

Conclusions

Conclusions should include the limitation of the research and the future research directions.

Reviewer 2 Report

GENERAL OVERVIEW

In this paper, the authors present a review of the existing works on the topics of Affinity Analysis, Bibliomining, and Technology Mining. For each one of these topics, it is presented a table specifying their respective major contribution, dataset used and implementation environment and evaluation.

WEAKNESSES

The topic of the paper is very interesting. However, in my opinion the paper has the following weaknesses:

- In my opinion this paper is not a systematic literature review as stated by the authors, but merely a summary of papers.

- I suggest reading the following references:

  • Pearl Brereton, Barbara A. Kitchenham, David Budgen, Mark Turner, and Mohamed Khalil. 2007. Lessons from applying the systematic literature review process within the software engineering domain. J. Syst. Softw. 80, 4 (2007),571–583. https://doi.org/10.1016/j.jss.2006.07.009
  • Barbara Kitchenham. 2004. Procedures for performing systematic reviews. Keele Univ. 33 (08 2004)

The procedure to perform systematic literature review is based on well-established guidelines for conducting systematic reviews, and is composed of the following steps:

— Analysis of related surveys: i.e, surveys, other systematic reviews or mapping studies) that, at least partially, cover the topics of Affinity Analysis, Bibliomining, and Technology Mining. The related surveys should be analyzed so that gaps and limitations must be identified and covered in the discussion.

— Definition of research questions: Based on the gaps identified in the previous step, the authors should define a set of research questions that form the objectives of the systematic review.

— Identification of studies: identify the data sources (e.g., search engines, online libraries) that are be used to identify relevant primary studies, i.e., the individual studies that contribute to the review.

— Study selection: This step refers to the application of inclusion and exclusion criteria on the primary studies found, as well as the quality assessment criteria used. The goal is to select the ones that agree with the goal and scope of the review.

— Data extraction and synthesis: this step extracts the relevant data (i.e., according to the research questions) from the selected primary studies. The information is then synthesized to allow answering the research questions.

- Which are the research questions for this review?

- The identification of the data sources is not complete. It is mentioned “others”. Google Scholar is used?

- The queries used to perform the search must be identified and mention that use some topics.

- The sentence in line 100 is not complete: “The rest of the paper is organized as follows: Section 2”. The complete structure of the paper should be described.

- How many researchers have done the selection of the most relevant papers? What happens if the researchers didn’t agree with the selection of a paper?  How many researchers are used in each phase?

- Table 2 shows the time slot distribution of the papers considered for affinity analysis. However, the final date is not shown (it is only 2019-Present). Which is the end date of the review?

- The previous also applies to “Table 3. Time slot distribution of the papers considered for bibliomining” and “Table 4. Time slot distribution of the papers considered for technology mining.”

- Which are the Open Issues and Challenges identified on the topics Affinity Analysis, Bibliomining, and Technology Mining?

- Which are the main threats to the validity of this work and the mitigation

Strategies?

As a general comment it is kindly recommended that native-English-speaker for linguistic improvements review this paper. Only some examples:

“Affinity analysis is the study of when objects exist together in other words correlated to each other.”

“In addition, such a study aids with up-selling and cross-selling, in addition to aiding with loyalty schemes, sales incentives, discount strategies, and even retail architecture.”

“Bibliomining is the use of measurable and example acknowledgment devices to a lot of information related to library frameworks to help dynamic or legitimize administrations.”

“Article is a poster, editorial, report, summery, book review.” Summery?

“We have finally selected 25 papers of affinity analysis, 15 papers of bibliomining and 7 papers in the domain of technology mining from various publishers like EEE, ACM, ScienceDirect, SpringerLink, Wiley and others.” EEE?

“For a decent comprehension of the innovation scene, designers frequently go to online assets.”

And many typos in the text.

In conclusion, the paper has many flaws and should have a profound revision.

Reviewer 3 Report

This paper reviews the existing works on Affinity Analysis, Bibliomining and Technology mining. However, I have many concerns as follows:

  1. The references list is too old. Only one reference dated one year ago, one dated two years ago and the rest are much older. Besides, some references are not presented adequately, ex: reference [27]. Considering how prolific the field is, authors have to extend their bibliography with significant, current references (up to 2022) and show that their problem is still relevant.
  2. There are major English language, grammar, spelling errors, and punctuation mistakes that should be thoroughly reviewed all over the manuscript, which makes the manuscript very poor language-wise. Ex: “In this article, we provides an up-to-date review which can serve ….”, “the readers can gain an proper overview of the ….”, “the quantity and data volume” redundancy of no added value , etc.
  3. “As a new field for database analysis, data mining, also referred to as….”, “the emerging field of data mining…” Is it possible in 2022 to consider “Data Mining” as a new emerging field? Especially when authors are referring to references that are more than 20 years ago?
  4. What is the motivation of this review? What is the research gap in the literature that this study is trying to fill-in?
  5. No related works section is included. The authors did not present any previous studies/reviews related to their presented work, which makes the references list miss many important references. What is unique about this review with respect to the other existing reviews? What is the different perspective adopted by the authors here?
  6. Research questions, that drive the paper, should be built in the introduction from an ongoing and pertinent bibliography (up to 2022). Identifying a research gap is a must.
  7. Many basic information and definitions are included, which makes the review like a tutorial rather than a research study.
  8. Authors did not provide any criticism, analysis, or evaluation to any of the mentioned studies in the review. Most of the content aggregates who did what, and what the other authors stated about their own results.
  9. No efforts detected to conduct any comparative study based on any pre-defined criteria for evaluation.
  10. How did the authors validate the “Implementation Environment and Evaluation” column stated in the 3 tables? What is included in this column is what has been stated in the other papers. Where is the analysis and discussion of the authors of this review?
  11. Answer your research question in the conclusions; what did we learn compared with current, significant research (up to 2022)?
  12. No future work is provided.
  13. The conclusion is very weak. What did you learn (that is not already known in the literature)? A discussion section is missing at the end of the manuscript to provide an overall evaluation to the current state of art, compared to the existing reviews in this field.
  14. Overall, a serious enhancement is required through a clarified detailed and deep analysis, discussions and comparisons with current, significant research (up to 2022).

Reviewer 4 Report

The author's did a great job by compiling this study, however, author claims in abstract  " This study reviews the existing works on Affinity Analysis, Bibliomining and Technology mining specifying their respective major contribution, used dataset and performance evaluation" has following limitations. 

  1. This study is missing most recent papers from the years 2020,  2021, and 2022.
  2. The paper inclusive criteria should be revised to add time frame of the papers in it.
  3. The author should consider combining table 2, 3, and 4 with a constant time interval for all three studies that is  Affinity Analysis, Bibliomining and Technology mining.
  4. The authors should provide the performance measuring criteria/benchmark's used in each papers of tables 5, 6, and 7 in a additional column. 

Round 2

Reviewer 1 Report

Thank you for considering the proposed revisions. 

Reviewer 2 Report

In this second version the quality of the paper has improved.

The authors have attending some of my comments and suggestions. However, there are some open issues, which we list below:

  • Abstract: “Three of the interesting fields of data mining are affinity analysis, bibliomining and technology mining.” should be “Three of the interesting fields of data mining are affinity analysis, bibliomining, and technology mining.”
  • Abstract: “Bibliomining is a combination of data mining, Bibliometrics and data warehouse.” should be : “Bibliomining is a combination of data mining, bibliometrics, and data warehousing.”
  • Abstract: “Technology mining is one of the current inquiry reasons which has been flaccid for bounty scientists in time association …” flaccid? bounty scientists? Please modify the language.
  • Abstract: “Finally, this article provides insight into major advances and outstanding challenges in the field of affinity analysis, bibliomining and technology mining.” should be “Finally, this article provides insights into major advances and outstanding challenges in the field of affinity analysis, bibliomining, and technology mining.”
  • Lines 82-83: “In technology management area, technology mining, or else its concision ‘tech mining’ is one of the current enquiries region which gradually developed day by day.” Improve the quality of this sentence.
  • Lines 358-361: “For finding massively lucrative itemsets with significant frequency proximity, a fondness utility value is used rather than high utility values in every transaction.” Fondness?? Please re-read the article [67] and modify the sentence accordingly.
  • Table 3, 4, and 5 are really interesting with the columns: Major Contribution, Data Set Used, Implementation Environment and Evaluation. In this new version the authors improve them adding a new column “Limitation”, which I agree. I only suggest modify the title in all tables to “Limitations”.
  • However, some entries should have more information. For example:
    • Nicholson et al. [28] “The precision of existing models can be improved”. How?
    • Hajek et al. [70] “This paper builds up a strategy for bibliomining, but the technique has numerous obstructions to its utilization” Which ones? Please give some examples.
    • Wu et al. [83] “The budget plan assignment is not limitless which is more composite and troublesome” Please specify.
  • The main weakness of this version is in section “8.1. Answers to the Research Questions”, because the authors did not explicitly answer some questions.
  • RQ3: What are the key bibliomining practices and technologies available for academic libraries? The authors only mention: “The answer to this research question can be obtained from the review done on bibliomining in Section 6. The summary of the associated studies along with their contribution, dataset, evaluation and limitation is presented in Table 4.” Please specify some key bibliomining practices and technologies.
  • RQ4: What are the existing techniques and approaches for technology mining? The authors only mention: “Similarly, the answer to this research question regarding technology mining can be obtained from the review provided in Section 7. The summary of the associated studies along with their contribution, dataset, evaluation and limitation is presented in Table 5.” Please give some examples of techniques and approaches.

In conclusion, the paper can be improved provided that the authors answer the above-mentioned questions and modify the paper according to the suggestions.

Reviewer 3 Report

Thank you for updating the manuscript. The content has been improved. However, I would strongly recommend to reconsider the "Implementation Environment and Evaluation" column in all the summary tables. Since Authors have not validated the outcomes of those experiments themselves, Authors cannot confirm the claims of other studies. Accordingly, it is recommended to focus on the provided facts in the presented studies rather than claims.

It is highly encouraged to review the English language throughout the manuscript.
